# Growth Performance and Fecal Microbiota of Dairy Calves Supplemented with Autochthonous Lactic Acid Bacteria as Probiotics in Mexican Western Family Dairy Farming

**DOI:** 10.3390/ani13182841

**Published:** 2023-09-07

**Authors:** José Martín Ruvalcaba-Gómez, Fernando Villaseñor-González, Mario Alfredo Espinosa-Martínez, Lorena Jacqueline Gómez-Godínez, Edith Rojas-Anaya, Zuamí Villagrán, Luis Miguel Anaya-Esparza, Germán Buendía-Rodríguez, Ramón Ignacio Arteaga-Garibay

**Affiliations:** 1Centro Nacional de Recursos Genéticos, Instituto Nacional de Investigaciones Forestales, Agrícolas y Pecuarias, Boulevard de la Biodiversidad 400, Tepatitlán de Morelos 47600, Jalisco, Mexico; gomez.lorena@inifap.gob.mx (L.J.G.-G.); rojas.edith@inifap.gob.mx (E.R.-A.); 2Campo Experimental Centro Altos de Jalisco, Instituto Nacional de Investigaciones Forestales, Agrícolas y Pecuarias, Av. Biodiversidad 2470, Tepatitlán de Morelos 47600, Jalisco, Mexico; villasenor.fernando@inifap.gob.mx; 3Centro Nacional de Investigación Disciplinaria en Fisiología y Mejoramiento Animal, Instituto Nacional de Investigaciones Forestales, Agrícolas y Pecuarias, Ajuchitlán Colón 76280, Querétaro, Mexico; espinosa.mario@inifap.gob.mx; 4Centro Universitario de los Altos, Universidad de Guadalajara, Av. Rafael Casillas Aceves 1200, Tepatitlán de Morelos 47600, Jalisco, Mexico; blanca.villagran@academicos.udg.mx (Z.V.); luis.aesparza@academicos.udg.mx (L.M.A.-E.); 5Sitio Experimental Hidalgo, Campo Experimental Valle de México, Instituto Nacional de Investigaciones Forestales, Agrícolas y Pecuarias, Carr. Pachuca-Tulancingo 104ª, Pachuca de Soto 42090, Hidalgo, Mexico; buendia.german@inifap.gob.mx

**Keywords:** *Lactobacillus*, PICRUSt, nutraceuticals, microbiota modulation, ion torrent

## Abstract

**Simple Summary:**

*Lactobacillus* is one of the bacterial genera frequently used as probiotics. Although probiotic effects are strain dependent, several *Lactobacillus* strains have been used to improve the growth rate in dairy calves, and the health or productivity parameters of dairy cows, mainly through the modulation of rumen microbiota. This study explains the effects observed on the growth parameters and fecal microbiota of dairy calves supplemented with three different *Lactobacillus*-based probiotic formulations supplied between days 5 and 19 after birth.

**Abstract:**

Probiotic supplementation in dairy cattle has achieved several beneficial effects (improved growth rate, immune response, and adequate ruminal microbiota). This study assessed the effects on the growth parameters and gut microbiota of newborn dairy calves supplemented with two *Lactobacillus*-based probiotics, individually (6BZ or 6BY) or their combination (6BZ + 6BY), administrated with the same concentration (1 × 10^9^ CFU/kg weight) at three times, between days 5 and 19 after birth. The control group consisted of probiotic-unsupplemented calves. Growth parameters were recorded weekly until eight weeks and at the calves’ ages of three, four, and five months. Fecal microbiota was described by high-throughput sequencing and bioinformatics. Although no significant effects were observed regarding daily weight and height gain among probiotic-supplemented and non-supplemented calves, correlation analysis showed that growth rate was maintained until month 5 through probiotic supplementation, mainly when the two-strain probiotics were supplied. Modulation effects on microbiota were observed in probiotic-supplemented calves, improving the Bacteroidota: Firmicutes and the Proteobacteria ratios. Functional prediction by PICRUSt also showed an increment in several pathways when the two-strain probiotic was supplemented. Therefore, using the three-administration scheme, the two-strain probiotic improved the growth rate and gut microbiota profile in newborn dairy calves. However, positive effects could be reached by applying more administrations of the probiotic during the first 20 days of a calf’s life.

## 1. Introduction

The intestinal microbiota has been recognized as a key organ in animals and humans due to its associated functions related to harvesting and absorbing nutrients involved in energy expenditure [1,2]. Specifically, the bovine gastrointestinal tract, considered sterile at animal birth, is rapidly colonized by bacteria, and it involves a very complex microbiota in response to several factors such as sex, genetics, and the exposome (feeding, environmental conditions, stress levels, among others) [3,4]; therefore, perturbations in gut microbiota have been related to the appearance or development of several pathologies or even to losses in metabolic and immunological abilities in the animal [3,5]. For this reason, microbiota modulation has become one of the main strategies to keep a healthy gut microbiota in animals to ensure its correct performance in obtaining energy from nutrients and modulating the immune response to improve tolerance and resistance to pathogenic agents [6,7]. 

In this context, it has been reported that correct colonization of gut microbiota at birth positively increases the success in the development of the immune system, contributing to keeping a good health status in animals [8]. Probiotics, which refer to the use of live microorganisms as a nutritional supplement to confer a health benefit to the host, have been one of the main microbiota modulation-based strategies to improve the health status and performance in several productive animal species, including dairy cows, alone or in combination with other strategies such as prebiotics [9,10,11]. Remarkably, dairy cow-associated rumen microbiota is mainly composed of members of the Firmicutes and Bacteroidetes phyla [12], strongly represented by *Clostridium*, *Lactococcus*, *Flavobacterium*, and *Prevotella* genera, and some other members of the Ruminococcaceae family [12]. The rumen-associated bacterial communities are acquired from the first days of life, before rumen development. These communities are mainly defined by the interactions of the newborn to environmental factors such as contact with the mother during the delivery, colostrum, and milk, and the establishment of the bacterial communities occurs the first 12 days after birth [13]. For this reason, it has been demonstrated that the intervention with bacteria-based probiotics is more efficient when performed in young calves [2,14].

Lactic acid bacteria (LAB), considered a natural resident of the gastrointestinal tract (GIT) and numerous fermented foods, have been extensively used as probiotics, mainly several members of the *Lactobacillus* genus, which was recently reclassified into 25 genera, due to its versatility in the use of substrates, colonization capacity, and beneficial effects [15,16,17,18]. This study aimed to assess the effects of using LAB-based probiotics on establishing GIT microbiota and growth parameters of calves allocated in Mexican western family dairy farming.

## 2. Materials and Methods

### 2.1. LAB Strains and Probiotic Preparation

Two LAB strains, previously isolated from the feces of dairy cows and characterized according to the FAO/WHO guidelines [19], were selected [20]. Strains were recovered according to the procedure described by Ruvalcaba et al. [21]. Strains were identified as *Lactobacillus* 6BZ and *Lactobacillus* 6BY. A pre-inoculum of each strain was prepared by inoculating 100 mL of MRS broth, followed by incubation at 37 °C for 24 h. Then, 24 h pre-inoculum was used for growth in a stirred-batch mode using a Biostat^®^ B Twin 5 L double-wallet, round-bottom glass bioreactor (Sartorius BBI Systems, Melsungen, DE). Temperature (35 °C), pH (6.5), and stirring (120 rpm) were automatically controlled. Next, 48 h bioreactor-produced cultures were mixed with sterile skim milk (10%, 1:1 *v/v*), distributed into 5 mL sterile serum bottles provided with two-leg Stoppers, and freeze-dried using a Labconco FreeZone^®^ 12 L Freeze Dry system equipped with a Stoppering Tray Dryer (Labconco, Kansas, MO, USA). The final concentration of each reconstituted probiotic dose was around 5 × 10^9^ CFU/mL. 

### 2.2. Animals, Treatments and Feeding

A total of 42 Holstein newborn calves were included in the study. Calves were allocated into three dairy family farms in the Los Altos de Jalisco region. At the age of 5 days, calves were aleatorily assigned to four treatments identified as the 6BZ group (*Lactobacillus* 6BZ supplemented; *n* = 8), 6BY group (*Lactobacillus* 6BY supplemented; *n* = 8), and 6BZ + 6BY group (supplemented with both 6BZ and 6BY strains; *n* = 17); and CG consisting of a control group (probiotic unsupplemented; *n* = 9). The calves’ feeding was based on a milk replacer (protein: 20%, fat: 19.5%, fiber: 0.9%) and then complemented with a calf starter concentrate that covered the nutrient requirements suggested by the Nutrient Requirements of Dairy Cattle of 2001 [22]. Water was offered at libitum from day two. Probiotics were reconstituted using the milk replacer and were supplemented at days 5, 12, and 19 after birth (1 × 10^9^ CFU/kg weight).

### 2.3. Growth Performance

From week two up to week eight of age, the calves’ weight was estimated weekly using a heart girth tape designed for Holstein`s calves [23]. The calves’ height was also recorded using a somatometric ruler (Nasco, Whitewater, WI, USA) [24]. Later, weight and height were recorded monthly up to month five. Data from weight and height were used to estimate daily weight gain and daily height gain. 

### 2.4. Feces Sampling and DNA Extraction

At days 1, 5, 15, 30, 45, and 60 after birth, 10 g of fecal samples was obtained from each calf. Samples were obtained aseptically before morning feeding, placed into sterile cryogenic vials (DNase-RNase free; Corning^®^, Glendale, AZ, USA), and brought to the laboratory. Samples corresponding to the same treatment, the same sampling day, and the same dairy farm were pooled, and 0.2 g of each pooled sample was used for DNA extraction. Metagenomic DNA was extracted and purified using the Quick-DNA™ Fecal/soil Microbe Miniprep system (Zymo Research, Irvine, CA, USA) according to the manufacturer´s instructions. DNA integrity was verified by electrophoresis in 1% agarose gel. DNA was stored at −20 °C until sequencing procedures.

### 2.5. Construction of the 16S rRNA Libraries and Sequencing Procedures

Libraries of 16S DNA were constructed based on PCR amplification of 7 of the 9 hypervariable regions of the 16S rDNA gene (V2, V3, V4, V6–V9). Amplification was performed in two independent reactions using the 16S Metagenomics™ system according to the manufacturer’s instructions (Thermo Fisher Scientific, Waltham, MA, USA) in a Verity™ thermal cycler (Thermo Fisher Scientific, Waltham, MA, USA). An equimolar mixture using the amplification products was prepared, and 50 nanograms were used to construct the 16S rDNA libraries with the Ion Plus Fragment Library commercial system and the Ion Xpress barcode adapters (Thermo Fisher Scientific). Library purification was carried out using the Agentcourt AMPure XP system according to the manufacturer’s instructions (Beckman Coulter, Brea, CA, USA) and quantified with a highly sensitive DNA commercial system and the Bioanalyzer 2100 (Agilent Technologies, Santa Clara, CA, USA). Library concentration was adjusted to 26 pM followed by PCR amplification of the PCR emulsion using a volume of 25 µL of the equimolar mixture for all samples (One-Touch 2, Thermo Fisher Scientific, Waltham, MA, USA) and enriched with the OneTouch Enrichment system (Thermo Fisher Scientific, Waltham, MA, USA). Sequencing was carried out using the Ion S5™ system (Thermo Fisher Scientific, Waltham, MA, USA).

### 2.6. Bioinformatics

Sequences were quality assessed and trimmed to remove low-quality regions using the Trimmomatic tool [25] and visualized in MultiQC [26]. Remotion of chimeric and low-quality sequences using the DADA2 module and ASV assignation using the SILVA v-132 database as a reference for the taxonomy assignation (16S rRNA gene sequences clustered at 99% of similarity) [27] were performed using the nf-core ampliseq pipeline v2.3.2 [28,29]. Alpha diversity (observed features, evenness, faith, and Shannon) was calculated using the QIIME2 [30] module and distances based on ASV abundance between samples (beta diversity) were measured using the method of “weighted UniFrac” and visualized using principal coordinate analysis (PCoA) plot. Relative abundance graphs and PCA graphics were generated using the abundance tables in the Origin v. 2022 software (OriginLab, Northampton, MA, USA). 

### 2.7. Functional Prediction

The software PICRUSt2 (v. 2.4.1) was used to conduct a deeper exploration of the possible metabolic mechanism associated with bacterial communities in the fecal samples, and obtain the functional profile [31]. The predictions were made by corresponding the marker gene data and the reference genomes in databases, including MetaCyc [32].

### 2.8. Statistical Analysis

The experimental design corresponded to a completely randomized design. Data corresponding to the growth performance were analyzed by REML ANOVA using the PROC MIXED procedure of SAS 9.4 v (SAS Institute Inc., Cary, NC, USA) for the repeated measures of weight and height obtained during the milk-feeding period (first eight weeks) and monthly records (0 to 5 months). Treatment, age, and treatment × age were considered fixed effects in the model, and the random effect of the block (production unit). The covariance structure was autoregressive. Significance was determined at *p* < 0.05. Pearson correlations were also calculated from weight and wither height data (α = 0.05).

## 3. Results

### 3.1. Growth Parameters

*Lactobacillus*-based probiotics were successfully supplemented to dairy calves, and the effects on growth parameters (weight and wither height) were recorded. Table 1 shows the weight and wither height of Holstein calves supplemented with probiotics during the milk-feeding period (first eight weeks of a calf’s life). Regarding the weight, no statistical differences were observed by probiotic treatment during the first eight weeks (*p* = 0.4308). Nonetheless, daily weight gain in the probiotic-supplemented calves was not significantly different (*p =* 0.456, Appendix A) in comparison with the non-supplemented calves. Regarding wither height scores, no statistical differences (*p* = 0.8233) were observed by probiotic supplementation. On the other hand, a significant effect was observed for both weight and wither height by age of calves (*p* < 0.001). Additionally, no interactions were observed between probiotic supplementation and age of calves for weight (*p* = 0.7198) and wither height (*p* = 0.4377) during the milk-feeding period.

Table 2 shows the weight and wither height of Holstein calves from birth to 5 months of age. In this context, during the first five months of a calf’s life, a significant effect was detected in the calf’s weight by treatment (*p* = 0.0480), month (*p* < 0.0001), and their interaction (treatment × age, *p* < 0.0001). In general, the calves treated with 6BZ (200.28 kg) and 6BY + 6BZ (202.00 kg) exhibited higher weights than the 6BY (184.66 kg) and control group (186.66 kg). On the other hand, no statistical effects were observed by probiotic supplementation (*p* = 0.3541) on wither height, but a significant effect associated with the calves’ age (*p* < 0.0001) was observed; moreover, no interactive effects (treatment group × age) were detected (*p* = 0.5826) for such parameter.

Aiming to corroborate the possible effects of probiotic supplementation on the maintenance of growth rate in calves, regression analysis of the body weight in week one, with the body weight recorded in the following seven weeks and months 3, 4, and 5 of age, was performed and the results are shown in Table 3. According to their *p*-values, correlations were considered null (0–0.01), weak (0.1–0.4), medium (0.4–0.6), or strong (0.6–1.0), when they were significant (*p* < 0.05). In the case of the control group, there was a positive correlation (*p* < 0.05) between the weight in week one and week 2. Subsequently, it was only significant in month 5. The groups of calves that received single-strain-based probiotics generally had a positive correlation only during the first 6 to 8 weeks of age, while in the group that received the two-strain-based probiotic, the correlation was maintained until five months of age (*p* < 0.05).

Similar results were observed when regression analysis was performed to wither height scores at different times during the study and the wither height values at the first week of age of the calves (Table 4). In general, positive and significant correlations (*p* < 0.05) were observed during the first six weeks for the calves in the control group. Meanwhile, positive and significant correlations were registered until week eight in the calves supplemented with the 6BZ strain probiotic. On the other hand, when the two-strain probiotic was supplemented, positive and statistically significant correlation values were observed up to month 3 of age.

### 3.2. Fecal Microbiota

The observed feature vector values differed between the probiotic-supplemented and non-supplemented calves (*p* = 0.018). In addition, based on the evenness vector, the abundance distribution of bacterial groups in fecal microbiota was different in the probiotic-supplemented calves compared to the non-supplemented group (*p* = 0.021). Both Shannon diversity index and faith diversity index values were higher in samples from the probiotic-supplemented calves (*p* < 0.05). Differences in relative abundances at different bacterial taxa were observed in feces from the probiotic-supplemented and non-supplemented dairy calves (Table 5).

In general, at least 19 Phyla in both types of samples were observed, with Firmicutes, Proteobacteria, and Bacteroidota being the most representative (Figure 1a). Bacteroidia, Bacilli, Clostridia, Negativicutes, Desulfovibrionia, Coriobacteriia, and Gammaproteobacteria were the main classes observed in the samples, together representing more than 90% of bacteria in the feces samples. Bacteroidia exhibited higher relative abundance in the probiotic-unsupplemented calves compared to the probiotic-supplemented groups. Bacteroidia’s relative abundance was higher in the non-supplemented calves in comparison to the probiotic-supplemented groups; meanwhile, Bacilli`s relative abundance was superior in samples from the probiotic-supplemented calves, mainly when the combination of probiotic strains (6BY + 6BZ) was used. The Clostridia proportion in feces was similar between the non-supplemented and 6BY + 6BZ-supplemented calves but remained higher when probiotic strains were used individually. Gammaproteobacteria’s relative abundance was higher in the non-supplemented calves during the first 45 days of evaluation compared to all probiotic-supplemented groups. However, at day 60, this abundance was similar to the individual strain groups but not to 6BY + 6BZ-supplemented calves that exhibited higher values.

Enterobacteriaceae was the most abundant family in calves´ feces at the beginning of the experiment but rapidly decreased, mainly in the probiotic-supplemented calves, and, at day 60, was practically undetectable, except in the 6BY-supplemented calves (Figure 1b). The Prevotellaceae family exhibited a significant increase in the four groups but, at day 60, remained higher in the 6BZ-supplemented calves, followed by the 6BY-supplemented associated samples and the non-supplemented calves; however, the 6BY + 6BZ-supplemented calves registered the lowest values for the relative abundance of this bacterial family. On the other hand, the Bacteroidaceae family remained under 3% of relative abundance at day 60, except in the 6BY-supplemented group that registered values closer to 6%. Additionally, on day 60, the Lachnospiraceae family exhibited higher relative abundance values in samples from the 6BZ- and 6BY-supplemented calves (13 and 17%, respectively) at day 60, meanwhile lower values were observed in feces from the non-supplemented and 6BY + 6BZ-supplemented calves (9 and 8%, respectively). 

The Muribaculaceae family was mainly observed at day 60 in feces from the non-supplemented calves (26% of relative abundance), followed by the 6BY + 6BZ-supplemented calves (10% of relative abundance). The Succinivibrionaceae family seemed to be increased by the use of the 6BY + 6BZ probiotic since relative abundance was higher in this group at day 60 (5%) compared to values observed in the rest of the groups (2% or less) as well as the Carnobacteriaceae family (3.5% of relative abundance in 6BY + 6BZ-supplemented calves vs. less than 1% in the rest of the groups), and the Acholeplasmataceae family (3.9% of relative abundance in 6BY + 6BZ-supplemented calves vs. less than 1% in the rest of the groups). The remaining bacterial families detected were found in low relative abundances (less than 1%). However, values were higher in feces from the 6BY + 6BZ-supplemented calves (11.5%) compared to the 6BZ-supplemented (7.2%), 6BY-supplemented (8.2%), or non-supplemented calves (6.9%), possibly indicating a greater bacterial diversity in that type of samples.

Finally, at genus level, the 6BY-based probiotic maintained the relative abundance of *Bacteroides* in the supplemented calves better than when the 6BZ-based or the 6BY + 6BZ-based probiotic was supplied. The *Alistipes* genus was also observed in all the samples, but reached its highest relative abundance at day 60 in samples from the 6BY- and 6BY + 6BZ-supplemented calves. The highest relative abundance of the *Prevotella* genus at day 60 was recorded in the samples from the 6BZ-supplemented calves, followed by samples from the 6BY-supplemented, non-supplemented, and 6BY + 6BZ-supplemented calves, respectively. 

On the other hand, the *Alloprevotella* genus exhibited superior relative abundances in samples from the probiotic-supplemented calves in comparison with the non-supplemented calves, being higher in the 6BZ-supplemented calves, followed by the 6BY-supplemented and 6BY + 6BZ-supplemented calves. The *Ruminococcus* genus was observed in all the samples but, at day 60, the relative abundance of this genus was higher in samples from the 6BY + 6BZ-supplemented calves. The relative abundance of the *Faecalibacterium* genus was also improved at day 60 when probiotics were supplemented, as well as observed for the *Pseudobutyrivibrio* genus. Differences among treatments, in terms of diversity at genus level, were corroborated through a three-component PCA graphic, explaining the 52% of the variance between samples (Figure 2).

### 3.3. Functional Prediction of Bacterial Communities

A total of 405 METACYC pathways were predicted to establish the functionality of fecal bacterial communities associated with probiotic-supplemented and non-supplemented dairy calves. The frequency of certain pathways was increased when probiotic supplementation was used, such as the L-lysine biosynthesis III, L-isoleucine biosynthesis II, starch degradation V, L-isoleucine biosynthesis IV, glycogen degradation I (bacterial), aromatic amino acid biosynthesis, coenzyme A biosynthesis I, glycolysis I (from glucose 6-phosphate), pyruvate fermentation to isobutano, glycolysis II (from fructose 6-phosphate), L-glutamate and L-glutamine biosynthesis, pyrimidine deoxyribonucleotides de novo biosynthesis III, tetrapyrrole biosynthesis II (from glycine), 4-deoxy-L-threo-hex-4-enopyranuronate degradation, L-methionine biosynthesis III, D-galacturonate degradation I, peptidoglycan biosynthesis IV (*Enterococcus faecium*), pyruvate fermentation to acetone, TCA cycle IV (2-oxoglutarate decarboxylase), glutaryl-CoA degradation, toluene degradation II (aerobic) (via 4-methylcatechol), and biotin biosynthesis II pathways (Figure 3).

## 4. Discussion

*Lactobacillus* is a bacterial genus commonly used in the probiotic formulation for humans and animals [17]. Particularly for animal production purposes, probiotic strains are generally isolated from animal feces and characterized to establish the safety and probiotic potential of the microbial strains, following well-validated protocols for this purpose, which has allowed strains with promising results for their use as probiotics to be obtained, as observed for various *Lactobacillus* strains obtained from buffalo or calf feces [33,34]. In this study, we evaluated the effects on dairy calves’ growth and fecal microbiota supplemented with three probiotic formulations using autochthonous lactic acid bacteria isolated from dairy cows´ feces. Two *Lactobacillus* strains were used as single-strain or two-strain probiotics, where no significant effects were observed on weight and wither height in dairy calves. Several studies have been conducted to assess the effects of different probiotic supplementation on the growth of dairy calves, reporting similar results, such as the study performed by Karamzadeh-Dehaghani et al. [35]. The authors in that study used a commercial probiotic containing dextrose, six different strains of lactic acid bacteria, and one *Bifidobacterium* strain, with no effects on body weight and growth parameters, including wither height after 28 days of evaluation. However, the authors reported improved diarrhea prevalence and some immune response indicators. Feed intake, body weight, and daily gain were neither improved by the concomitant use of a *Bacillus*-based probiotic and nucleotides derived from yeast supplementation through milk replacer in calves, but nucleotide supplementation seemed to reduce the *Lactobacillus* concentration in feces [36]. Nonetheless, some probiotic strains induce increases in calf daily weight gain, as reported by Jiang et al. [37] using a *Lactiplantibacillus plantarum* 299v strain as probiotic Holstein calves. They mentioned that the probiotic significantly increased the feed starter intake, as well as the average daily gain in the supplemented calves, at the time, reduced the diarrhea incidence, and increased the glucose, IgG, IgA, Interferon-gamma, and soluble CD4+ concentrations in plasma. 

In this study, results suggested that probiotic supplementation significatively improved the maintenance weight gain of calves during their first five months of life, mainly after the milk-feeding period (first eight weeks of life), possibly by the microbiota modulation in calves that induces a medium- and long-term effect on the growth parameters, mainly when the two-strain probiotic was used [38]. Effects on feed efficiency, body weight gain, and reduced incidence of diarrhea have been commonly assessed when LAB-based probiotic supplementation is implemented in calves, but the observed results may depend on the bacterial strains. Some meta-analyses have intended to summarize the main effects of LAB-based probiotic supplementation. Through a meta-analysis of randomized controlled trials regarding LAB supplementation as a probiotic to young calves, including nine studies, Signorini et al. [39] observed that only when multi-strain probiotics were used the diarrhea incidence in the calves was reduced. Meanwhile, another meta-analysis published by Wang et al. [40], which included 49 studies of probiotic supplementation to pre-weaning dairy calves, reported that probiotics improved the growth performance but decreased digestibility and feed efficiency by increasing the dry matter intake and concluded that effects are indexed to the probiotic strain, supplementation dosage, and methods. On the other hand, Dehghan et al. [41] mentioned that, after analyzing the results of eight articles related to the probiotic supplementation to dairy calves, no significant effects on dry matter intake were observed as well as for feed efficiency; and Frizzo et al. [42] indicated, as a result of the meta-analysis of 21 articles and 14 studies, that growth of calves did not change when a LAB-based probiotic was supplemented in whole milk; however, beneficial effects were achieved when they were added to a milk replacer. The authors also highlighted that the number of supplemented strains seemed not to affect the results, but conclusions could be related to the number of calves included in each experiment. 

In general, microbiota modulation is the main wanted effect when probiotics are used [43]. In this study, we proposed an administration scheme that only considered the administration of the probiotic, after the colostrum at day five after birth, on three occasions with a time between administrations of seven days. Following the administration scheme, we observed differences in fecal microbiota composition and structure as an indicator of changes in intestinal microbiota which, eventually, will promote the rumen microbiota establishment in the calves. Differences in the relative abundance of the different taxa were observed since day 15 and generally maintained until day 60.

Regarding phylum, the structure of the fecal microbiota observed in the samples from probiotic-supplemented calves was comparable to microbiota reported for healthy calves [44], which is mainly characterized by the maintenance of the proportion 1:1 or closed to between phyla Firmicutes and Bacteroidota [45] that are considered as a biomarker for metabolic potential of the gut microbiota [41]. Similar results were reported by Chang et al. [46] after the calves’ supplementation with galacto-oligosaccharides, which facilitated the increase in the relative abundance of beneficial bacterial in the rumen, promoting the growth of the calves and reducing the incidence of diarrhea. Additionally, Actinobacteria and Proteobacteria phyla tended to increase in supplemented calves compared to the control. Although these phyla used to predominate in fecal microbiota of diarrheic calves [47], the observed values of relative abundance remained lower than those reported to cause dysbiosis in calves, particularly considering the Proteobacteria: Firmicutes + Bacteroidetes ratio, as reported previously [48] in beef cattle. On the other hand, Bacteroidia and Clostridia were the most abundant classes in all the treatments. It has been reported that Clostridia could be considered a marker of gastrointestinal dysbiosis [44]; in contrast, Bacilli and Gammaproteobacteria have been observed as predominant classes in healthy calves [43]; moreover, abundance of Bacteroidia plays an essential role in nutrition mainly for its contribution to carbon degradation [49].

Under day 60 after birth, the leading bacterial families in samples from the probiotic-supplemented calves were Prevotellaceae, Muribaculaceae, Succinivibrionaceae, and Acholeplasmataceae. It has been reported that the Prevotellaceae family usually increases with age in neonatal dairy calves, particularly after the first eight weeks of life [50], and is considered an essential family involved in carbohydrate degradation such as starch, xylan, pectin, and hemicellulose to produce propionate, succinate, and acetate [46,51], as well as in protein degradation to obtain peptides in the rumen [52]. The abundance of this family of microorganisms helps maintain calves’ normal digestive function [49]. Prevotellaceae has been reported to be the most abundant family in fecal microbiota from Holstein calves fed with *Lacticaseibacillus rhamnosus* CG during the preweaning stage [49]. Additionally, the presence of Muribaculaceae family members has been related to a lower incidence of diarrhea in neonatal calves [53]; besides, they can degrade complex polysaccharides and potentially produce short-chain fatty acids such as acetate, propionate, and butyrate [54]. According to Kodithuwakku et al. [55], Muribaculaceae in the rumen contribute to efficient feed conversion into energy sources for milk production in dairy cows. Furthermore, the increase in the Succinivibrionaceae members’ family has been reported as promoters for establishing the early rumen microbiota of young calves [56]. Similar trends were observed in the Acholeplasmataceae family, which was previously reported as part of the microbial taxon reported for rumen fluid, maintaining relative abundances of ≈5% in adult cows [57].

Regarding Genus taxa, *Bacteroides*, *Alistipes*, *Prevotella*, *Alloprevotella*, *Faecalibacterium*, and *Pseudobutyrivibrio* were the genera that exhibited the most notable differences in terms of relative abundance, depending on the study group at the age of 60 days. *The Bacteroides* genus is essential in absorbing amino acids in the intestine [49]. Furthermore, *Alistipes* contain a polysaccharide-degrading enzyme that facilitates the complex oligosaccharides degradation, and it has been reported with higher prevalence in the development of mature rumen of dairy calves [58]. A high abundance of *Bacteroides* and *Alistipes* in the feces of *Lactiplantibacillus plantarum* 299v-supplemented preweaning calves has been reported compared to non-supplemented calves [37]. *Prevotella* has been reported as a predominant genus in newborn calves during the first two weeks of age [50]; moreover, it is a key rumen microbial genus in calves [46], recognized as responsible for maintaining the normal digestion process of the calf’s rumen [49] and protein degradation [50]. The *Prevotella* genus was frequently identified in the feces of beef cattle fed high-energy diets as a component of the core microbiome in the absence of probiotics [59,60]. Furthermore, it has been reported that a high abundance of *Prevotella* and *Alloprevotella* in dairy calves could exert a probiotic effect by reducing the incidence of diarrhea in the early stage [53]. On the other hand, *the Faecalibacterium* genus increased in feces from probiotic-supplemented calves compared to the control group. It has been reported that the fecal microbiota of pre-weaned dairy calves was dominated by microorganisms of this genus, which exhibited a lower incidence of diarrhea during the first four weeks [58]. Moreover, this genus has been associated with butyrate production, which promotes a greater weight gain during the first weeks of calves’ lives due to butyrate enhancing the integrity of the intestinal epithelial barrier that may reduce susceptibility to intestinal infections in calves [61]. Virginio et al. [62] reported that both genera *Allopevotella* and *Faecalibacterium* increased in abundance in dairy calves supplemented with β-glucans until week eight. Finally, *Pseudobutyrivibrio* increased its relative abundance in feces from probiotic-supplemented calves. Some *Faecalibacterium* species have been isolated from various ruminants, and reports indicate that they can use complex polysaccharides as substrates for growth, including xylan and hemicellulose. However, the *Pseudobutyrivibrio* proportion has been reported to be decreased in the rumen of older calves [63].

The current results suggest that probiotic supplementation regulates the bacterial community composition in the intestine of calves, particularly in 6BY + 6BZ-supplemented calves (mainly in terms of diversity at the genus level) that may promote the absorption of nutrients in diets [37]. It has been reported that feeding lactic acid bacteria preparations for one day affects the microbial species composition of the feces of calves through *β* diversity data analysis using multivariate statistical tools such as principal component analysis [64]. Additionally, Fan et al. [65] reported that the *β* diversity of fecal microbiota of Holstein dairy calves was influenced by the supplementation with milk replaced with ethoxyquin, improving early rumen microbial development; this suggests that the interactions among different bacteria genera might play an essential role in maintaining intestinal homeostasis in the gut of newborn calves; moreover, the major impact on microbiota composition and diversity is directly related to the combined probiotic supplementation [65]. The changes observed in the structure of the bacterial community from feces were reflected in the predicted functional profile of samples. At least 18 predicted MetaCyc pathways showed incremented frequencies compared with non-supplemented calves. Samples from calves that were supplemented with the single-strain probiotic exhibited changes in the frequency of the selected pathways at day 15, but it seemed those frequency levels were not maintained at day 30 and later; on the other hand, when the 6BZ + 6BY-based probiotic was supplemented, the feces samples exhibited a sustained increment in the frequency of these predicted pathways.

## 5. Conclusions

This study demonstrated that using two *Lactobacillus* strains for calves’ supplementation, considering three administrations during the two weeks after colostrum, resulted in higher weights after the milk-feeding period and at least during the first five months of life of supplemented calves mainly when the combined two strains were used. This could be related to the gut microbiota modulation, expressed as changes in composition and structure of the fecal microbiota of the probiotic-supplemented calves, promoting a balance between the most representative bacterial phyla. Nonetheless, exploring a different administration scheme that considers a more extended exposition of calves to the probiotic would be essential to improve the daily weight and height gain and increase, to a greater extent, the relative abundance of groups of microorganisms considered as probiotics in the rumen.

## Figures and Tables

**Figure 1 animals-13-02841-f001:**
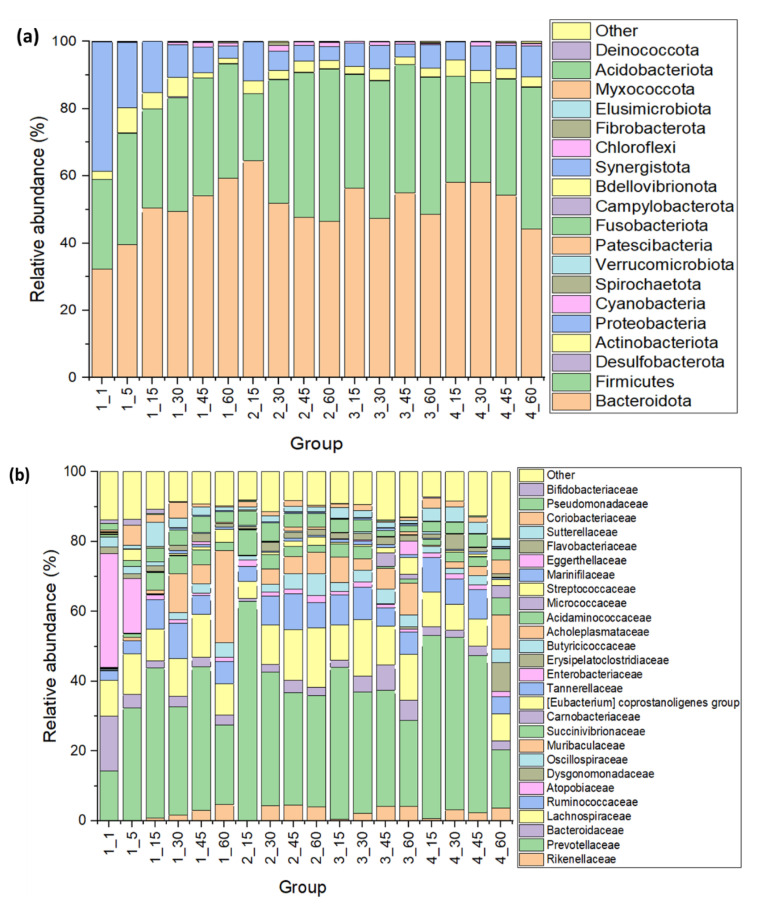
Relative abundance of main bacterial (**a**) Phyla and (**b**) families in feces from probiotic-supplemented and non-supplemented dairy calves’ groups: (1: non-supplemented calves; 2: 6BZ-strain probiotic; 6BY-strain probiotic, 6BZ + 6BY probiotic) at different times after birth (1, 5, 15, 30, 45, and 60 days).

**Figure 2 animals-13-02841-f002:**
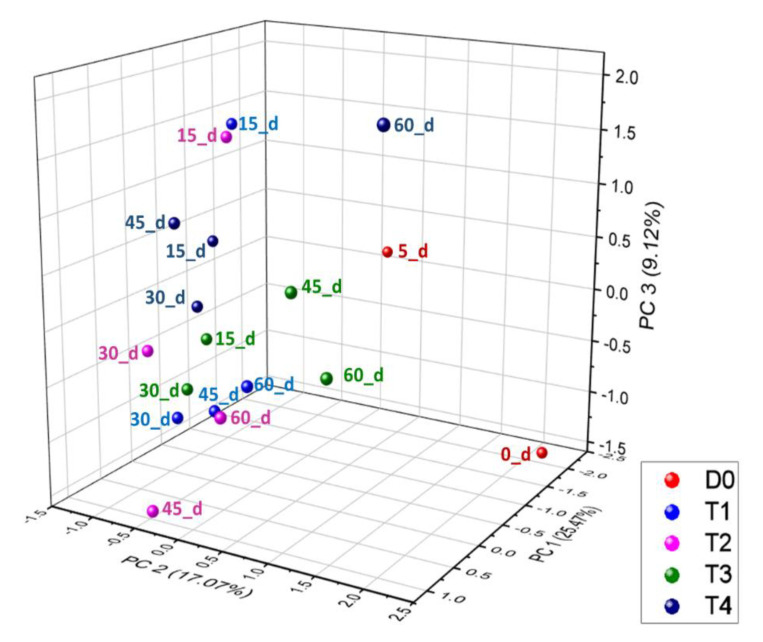
Principal component analysis graphic calculated from relative abundance of bacterial genera in feces samples from probiotic-supplemented and non-supplemented dairy calves’ groups: (T1: non-supplemented calves; T2: 6BZ-strain probiotic; T3: 6BY-strain probiotic, and T4: 6BZ + 6BY probiotic) at different times after birth (1, 5, 15, 30, 45, and 60 days).

**Figure 3 animals-13-02841-f003:**
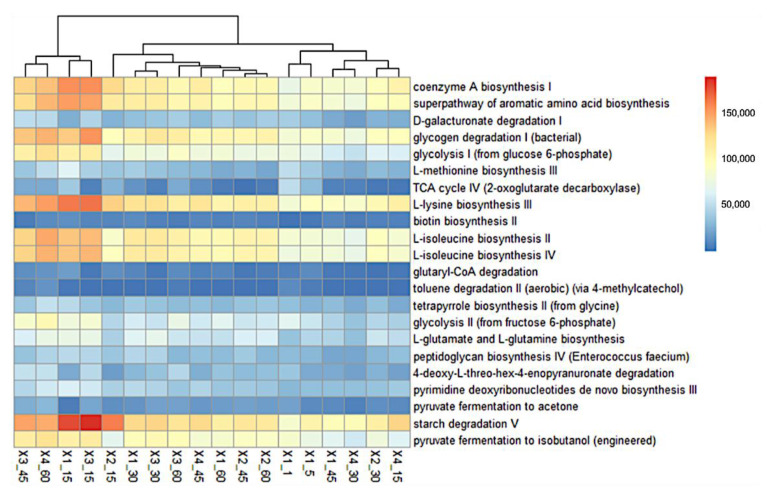
Heatmap representation of the main PICRUSt-predicted MetaCyc pathways incremented in feces from probiotic-supplemented calves. X1: non-supplemented calves, X2: 6BZ-supplemented group, X3: 6BY-supplemented group, X4: 6BZ + 6BY-supplemented group at different times after birth (1, 5, 15, 30, 45, and 60 days).

**Table 1 animals-13-02841-t001:** Body weight and wither height of Holstein calves during milk-feeding period.

Parameter/Age (Week)	Calves’ Groups (Teatment)				Significance		
	Control	6BZ	6BY	6BY + 6BZ	T	A	T × A
Body weight (kg)							
0	39.55 ± 0.37	42.12 ± 0.76	40.62 ± 1.22	40.23 ± 0.64	0.4308	<0.0001	0.7198
1	41.11 ± 0.78	42.87 ± 1.02	42.50 ± 1.42	42.23 ± 0.92			
2	43.33 ± 1.05	45.62 ± 1.06	44.00 ± 1.80	44.29 ± 1.18			
3	47.11 ± 1.33	48.87 ± 1.43	45.75 ± 2.20	47.58 ± 1.15			
4	50.88 ± 1.52	54.75 ± 1.67	49.25 ± 2.45	52.76 ± 1.47			
5	54.88 ± 1.78	59.50 ± 2.28	55.00 ± 3.21	57.47 ± 1.67			
6	61.11 ± 2.07	64.12 ± 2.18	58.87 ± 3.51	62.82 ± 2.01			
7	67.77 ± 3.13	71.00 ± 2.64	66.25 ± 2.51	69.82 ± 2.51			
8	72.55 ± 3.01	78.25 ± 3.78	73.12 ± 3.30	77.52 ± 2.92			
Wither height (cm)							
0	78.66 ± 0.91	80.50 ± 0.53	80.50 ± 0.90	78.70 ± 0.98	0.8233	<0.0001	0.4377
1	80.11 ± 1.05	80.62 ± 0.94	81.12 ± 1.07	79.58 ± 0.74			
2	81.88 ± 1.00	82.50 ± 0.88	82.50 ± 0.80	81.47 ± 0.73			
3	84.22 ± 0.79	84.25 ± 0.95	84.12 ± 0.85	83.41 ± 0.71			
4	85.77 ± 0.74	86.50 ± 0.84	84.87 ± 0.93	85.70 ± 0.64			
5	87.88 ± 0.77	87.87 ± 1.04	86.75 ± 1.17	87.52 ± 0.79			
6	89.22 ± 0.86	90.62 ± 1.01	88.50 ± 1.21	89.23 ± 0.82			
7	91.88 ± 1.47	92.00 ± 1.19	89.25 ± 1.91	91.11 ±0.75			
8	93.77 ± 0.87	93.50 ± 1.23	92.25 ± 1.91	93.29 ± 0.98			

Average value ± standard error. T: treatment group; A: age in weeks, T × A: treatment group × age.

**Table 2 animals-13-02841-t002:** Body weight and wither height of Holstein’s calves from birth to 5 months of age.

Parameter/Age (Months)	Calves’ Groups (Teatment)				Significance		
	Control	6BZ	6BY	6BY + 6BZ	T	A	T × A
Body weight (kg)							
0	39.55 ± 0.78	42.12 ± 1.02	40.62 ± 1.22	40.23 ± 0.64	0.0480	<0.0001	<0.0001
1	101.11 ± 4.48	107.8 ± 4.20	99.50 ± 2.85	96.61 ± 2.61			
2	123.55 ± 5.77	135.4 ± 3.16	130.37 ± 5.02	119.94 ± 3.46			
3	145.66 ± 4.98	155.28 ± 3.16	157.37 ± 7.97	152.33 ± 5.15			
4	166.55 ± 4.92	180.14 ± 4.65	195.87 ± 17.63	159.66 ± 4.44			
5	186.66 ± 4.57	200.28 ± 5.97	202.00 ± 12.73	184.66 ± 5.90			
Wither height (cm)							
0	78.66 ± 0.91	80.50 ± 0.53	80.50 ± 0.90	78.70 ± 0.98	0.3541	<0.0001	0.5826
1	98.33 ± 1.01	96.42 ± 2.10	97.62 ± 1.22	95.38 ± 0.74			
2	102.00 ± 1.01	101.42 ± 1.19	102.62 ± 1.71	95.38 ± 0.63			
3	107.88 ± 0.88	107.57 ± 1.26	108.00 ± 2.35	100.72 ± 0.74			
4	111.33 ± 0.84	111.28 ± 1.40	113.00 ± 2.14	107.44 ± 0.76			
5	116.55 ± 1.00	118.28 ± 1.24	117.50 ± 1.76	110.38 ± 0.66			

Average value ± standard error. T: treatment group; A: age in months, T × A: treatment group × age.

**Table 3 animals-13-02841-t003:** Pearson’s correlation coefficients and probability values (*p* values), among first week and different ages, for the weight of both probiotic-supplemented and non-supplemented calves during the study.

		Age									
		Weeks							Months		
Treatment		2	3	4	5	6	7	8	3	4	5
Control	P	0.8955	0.590	0.590	0.514	0.380	0.382	0.498	0.528	0.601	0.681
	*p*	0.001	0.094	0.094	0.156	0.313	0.310	0.172	0.143	0.086	0.04
6BZ	P	0.889	0.881	0.872	0.735	0.724	0.695	0.858	0.255	0.217	0.149
	*p*	0.003	0.003	0.004	0.037	0.042	0.055	0.006	0.542	0.606	0.724
6BY	P	0.907	0.655	0.766	0.782	0.743	0.612	0.609	0.190	0.519	0.277
	*p*	0.001	0.077	0.026	0.021	0.034	0.106	0.109	0.652	0.187	0.507
6BY + 6BZ	P	0.914	0.813	0.631	0.633	0.652	0.488	0.526	0.579	0.653	0.801
	*p*	<0.0001	<0.0001	0.006	0.006	0.004	0.046	0.03	0.014	0.004	0.0001

P = Pearsons’ correlation coefficients; *p* = probability values. *p* values ≤ 0.05 are statistically significant.

**Table 4 animals-13-02841-t004:** Pearson’s´ correlation coefficients and probability values (*p* values), among first week and different ages, for the wither height of both probiotic-supplemented and non-supplemented calves during the study.

		Age									
		Weeks							Months		
Treatment		2	3	4	5	6	7	8	3	4	5
Control	P	0.939	0.639	0.782	0.579	0.741	0.492	0.481	0.547	0.478	0.465
	*p*	0.000	0.064	0.013	0.102	0.022	0.178	0.190	0.127	0.193	0.207
6BZ	P	0.879	0.231	0.269	0.455	0.413	0.428	0.581	0.502	0.689	0.527
	*p*	0.004	0.583	0.520	0.257	0.309	0.290	0.131	0.205	0.059	0.180
6BY	P	0.822	0.725	0.795	0.683	0.872	0.712	0.786	0.358	0.320	0.430
	*p*	0.012	0.042	0.018	0.062	0.005	0.047	0.021	0.384	0.440	0.287
6BY + 6BZ	P	0.886	0.820	0.664	0.661	0.601	0.541	0.626	0.538	0.472	0.457
	*p*	<0.000	<0.000	0.004	0.004	0.011	0.025	0.007	0.026	0.056	0.065

P = Pearsons’ correlation coefficients; *p* = probability values. *p* values ≤ 0.05 are statistically significant.

**Table 5 animals-13-02841-t005:** Relative abundance of main bacterial taxa that exhibited differences in feces from *Lactobacillus*-based probiotic-supplemented or non-supplemented dairy calves.

	Treatment			
	Average Relative Abundance (%, at Day 60)			
	Control	6BZ	6BY	6BY + 6BZ
Phylum				
Bacteroidota	59.3	46.5	48.6	44.1
Firmicutes	34.0	45.3	40.6	42.2
Actinobacteriota	1.7	2.5	2.6	3.1
Proteobacteria	3.6	4.1	7.0	9.1
Class				
Bacteroidia	59.3	46.5	48.6	44.1
Bacilli	3.6	3.5	6.5	12.3
Clostridia	26.9	37.5	32.1	26
Negativicutes	3.3	4.0	1.8	3.2
Desulfovibrionia	0.0	0.0	0.1	0.1
Coriobacteriia	1.6	2.4	1.7	1.7
Gammaproteobacteria	3.4	3.9	6.6	8.7
Family				
Enterobacteriaceae	0.0	0.0	3.85	0.09
Prevotellaceae	22.7	31.9	24.8	16.7
Bacteroidaceae	2.9	2.3	5.6	2.6
Lachnospiraceae	8.9	17.1	13.2	7.7
Muribaculaceae	26.4	6.1	9.1	9.8
Succinivibrionaceae	2.3	2.1	1.2	5.0
Carnobacteriaceae	0.0	0.1	1.3	3.5
Acholeplasmataceae	0.3	0.8	0.8	3.9
Genus				
*Bacteroides*	2.9	2.3	5.6	2.6
*Alistipes*	0.3	0.1	0.6	0.5
*Prevotella*	17.7	21.7	18.5	10.3
*Alloprevotella*	3.7	9.1	5.8	5.0
*Ruminococcus*	1.7	2.0	2.5	2.2
*Faecalibacterium*	0.7	1.0	1.1	1.1
*Pseudobutyrivibrio*	0.07	0.34	0.23	0.15

## Data Availability

The derived 16S rRNA gene sequences are available at the NCBI under the Bioproject ID: PRJNA991611.

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
