# Peer review of "Growth Performance and Fecal Microbiota of Dairy Calves Supplemented with Autochthonous Lactic Acid Bacteria as Probiotics in Mexican Western Family Dairy Farming"

_animals, 2023, doi:10.3390/ani13182841_

Round 1

Reviewer 1 Report

GENERAL COMMENTS

The nomenclature of lactic acid bacteria has been modified and should be updated throughout the manuscript.

Lactic acid bacteria tend to modify the intestinal microbiota, not the rumen, especially in calves.

SPECIFIC COMMENTS

Abstract: It is not clear if there are three types of LAB-based probiotic formulations and two doses or two LAB-based probiotics with three doses.

"...Although no significant effects (p > 0.05) were observed in terms of daily weight and height gain among..." and then "...using the tree-doses scheme, the two-strain probiotic improved the growth rate and...". Were there or were there no effects on growth performance?

Material and Methods

Animals, treatments and feeding: Why was the number of animals included in each experimental group so different?

Line 108: Missing close parenthesis. In the abstract, the authors talk about giving three doses, but using different doses was not mentioned here.

Line 110: Calf weight measurements were widely spaced in time, which can make it difficult to see differences in specific periods.

Line 115: The authors carried out three measurements of the calves´ weight throughout the study but took 6 fecal samples for microbiological analysis. Why?

Line 157: The experimental design is described in a very confusing and changing way. From what I can reconstruct, it looks like it was a factorial design with probiotic formulations and dosage as the main factors. Additionally, measurements of weight and intestinal microbiota were performed at different moments of the study, which implies the existence of repeated measurements. None of this is visualized in the statistical analysis. Was it confirmed that the frequency distribution of the growth performance was normal and homoscedastic?

Results:

Line 164: "...Regarding the daily weight gain, the analysis of variance, including the production unit as a block in the model, did..." Block? the existence of blocks was not described.

Line 167: please, provide the exact P-value.

Line 168: "...For daily height gain, only a tendency to be different between treatments was observed,,," Higher in the control group.

Line 173: Here the existence of repeated measures is mentioned. However, the analysis has to be performed considering the repeated measures; two different analyzes cannot be carried out for the same variable and design.

Line 179: This is fine, but this is desirable and not attributable to the probiotic.

Line 183: "...Aimed to assess possible effects of probiotic supplementation on the maintenance of growth rate in calves, regression analysis of the body weight..." A correlation analysis was mentioned previously.

The authors should further explain the results in Table 4 as well as the implications they believe these results have. As far as I can understand, the results of the Pearson correlation indicate that the monthly weight of the calves is correlated with the initial one, that is, the heaviest at the beginning were the heaviest at 5 months. What is the significance of these results? Is the probiotic effect verified when this correlation remains high?

Line 219: How did they make this comparison? based on what statistical technique?

Figure 2: Principal components analysis was not mentioned in the Material and Methods section.

Discussion:

Line 316: the results are repeated.

Lines 319 to 329: There are meta-analyses that summarize the findings of different similar studies. Maybe it could be based on them and not comparing results with particular articles.

Lines 333-335: This is not correct, the fact that there is a correlation with the initial weight is not an indicator that the weight gain is maintained. In that case, I could have statistically analyzed the daily, monthly, or weekly gain and see if it changed.

Line 341: The effect of probiotics in calves is given by the modification, essentially, of the intestinal microbiota.

Line 344: The health events were measured? how did the probiotic behave?

Conclusions

Line 444: Dosages or administrations?

Line 445: This was not proven, the statistical method used does not allow us to conclude this.

Reviewer 2 Report

The manuscript presents a well-defined objective in the abstract and introduction. The introduction justifies the reasons why the investigators decided to carry out this study by introducing recently published work.

Materials and Methods

The experimental design is missing to understand the trial.

Why is the number of animals in the 6BZ+6BY group (n=17) double that of the other groups (n=8)?

Line 106 - you should state which NRC you are referring to (date is missing).

The composition of the feed, in this case milk, should be stated in the paper.

L110 - should replace calculated weight with estimated weight.

Results

In Table 1, the authors state that for cattle growth there was a tendency for growth to decrease when probiotics were added. I am not sure if they can say that there is a trend as there was no replication of results, i.e. in this trial, it was not statistically different, as with weight.

Table 2 legend - "statistocal"

Table 3 - indicates the statistical differences, but does not indicate the p-value for each parameter.

In my opinion this table would make sense, if they compared in each month the results of each probiotic. It does not make sense to use the averages of all animals in all groups and compare over 5 months. It is natural that "natural" growth occurs even without the influence of probiotics.

Table 4

It is not necessary to put the "p=" inside the table

Discussion:

What the authors mean by "whiter height"

Overall the discussion focuses on the microbiological analyses of the calves' faeces, where they obtained better results. However, the correlations presented in the results are sometimes significant, but they are weak correlations, close to 0.5 or even lower, but this part is a bit omitted in the discussion, it should be referenced and justified why these correlations are weak, especially in height growth.

General notes:

Structurally some commas, brackets, etc. are missing.

For example:

L108 -" (...)days 5, 12 and 19 after birth (1x10 108 9 CFU/kg weight." The final parenthesis is missing.

P- value - sometimes appears in italics sometimes not.

The conclusion reflects the work, answering the proposed objective, indicating what could be improved in the future to improve the growth and weight of the animals.

Round 2

Reviewer 1 Report

GENERAL COMMENTS

The nomenclature of lactic acid bacteria has changed, and authors must adhere to it. It is not a matter of preferring one or another way of naming bacteria but of how it should be done.

Please, provide the exact P-values.

SPECIFIC COMMENTS

Line 32: The authors keep mentioning that three doses of each were used, which is confusing. If only one dose was used for all three LAB treatments, then say exactly that.

Line 162: The same problems were detected regarding the definition of the type of experimental design used. As I understand, the authors used a repeated measures model (one treatment with four levels and experimental units sampled at different times throughout the study). On the other hand, it was not completely random because the number of animals included in each group was very different, assuming a non-random assignment.

Line 175: There were not differences (P=0.456).

It is not clear what was done with the data depicted in Figures 1 A and B. Please explain more clearly.

Lines 201-203: I disagree with the author's response in the review note and in the way in which they intend to conclude about the effect of probiotics on animal performance. It is not correct to conclude that the treated animals maintained their growth through the correlation with birth weight.

Discussion: In my previous review, I suggested using meta-analyses carried out on the topic to discuss the results. In the review note, the authors misinterpret the results and scope of this type of study. Meta-analyses do not generalize results as they allow the identification of study factors that may explain the reasons why the results between studies differ. However, it was only a suggestion to facilitate and broaden the discussion.

Do not repeat results in the discussion section.
